# Predicting the Geographical Distribution Shift of Medicinal Plants in South Africa Due to Climate Change

Thulani Tshabalala [1,*], Onisimo Mutanga [1] and Elfatih M. Abdel-Rahman [2]

1 School of Agricultural, Earth and Environmental Sciences, University of KwaZulu-Natal Pietermaritzburg, Private Bag X01, Scottsville 3209, South Africa
2 Data Management, Modelling and Geo-Information Unit, International Centre of Insect Physiology and Ecology (icipe), Nairobi 00100, Kenya
* Correspondence: tshabalalat1@ukzn.ac.za

**Abstract:** There has been a recent rise in the number of medicinal plant users in Southern Africa, with approximately a million users reported to utilize these plants for various health conditions. Unfortunately, some of these plants are reportedly endangered and facing extinction due to harvesting pressure. In addition, climate change is likely to negatively affect the geographical distribution of these medicinal plants. In the current study, future greenhouse gas emission scenarios of the representative concentration pathways, RCP2.6 and RCP8.5, for future projections to 2050 and 2080 were used to simulate the effect of climate change on three medicinal plants' (*Aloe ferox*, *Bowiea volubilis*, and *Dioscorea elephantipes*) distribution in South Africa. We studied these plant species as the International Union for Conservation of Nature stated that *A. ferox* is currently of least concern in South Africa, while *B. volubilis* and *D. elephantipes* are categorised as declining and vulnerable, respectively. Specifically, we utilised a species distribution model (i.e., the maximum entropy: MaxEnt) to investigate the effect of climate change on the future spatial distribution of medicinal plants in South Africa. In 2050 and 2080, under both RCP scenarios, the suitable habitat of the studied plant species will reduce in the country's northern parts. Specifically, the habitat for *D. elephantipes* will totally disappear in the country's northern parts. However, there will be slight additions of suitable habitats for the species in the country's southern parts. Model validation indicated that the area under curve (AUC) for *A. ferox* was $0.924 \pm 0.004$, while for *B. volubilis* and *D. elephantipes* it was $0.884 \pm 0.050$ and $0.944 \pm 0.030$, respectively. Using the results from this study, there is a need for the long-term in situ and ex situ conservation of these medicinal plants. The results of the present study could guide the development of effective and efficient policies and strategies for managing and conserving medicinal plants in South Africa.

**Keywords:** *Aloe ferox*; *Bowiea volubilis*; *Dioscorea elephantipes*; MaxEnt; species distribution modelling; species vulnerability

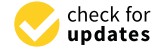

## 1. Introduction

Medicinal plants are an essential component of health care for most people living in developing countries, with up to 95% of the population utilising at least one form of their materials [1]. There are approximately 100 million users of medicinal plants in Southern Africa, amounting to 700,000 tonnes of plant material being harvested annually [2]. There has been a rise in the use of medicinal plants, and they are mostly preferred to conventional medicines because of their affordability, availability, and fewer side effects [3]. These plants have the most significant potential to benefit humankind, especially those living in third world countries where poverty, poor health, and high unemployment rates are prevalent. In addition, these plants are a resource for poverty alleviation. For instance in South Africa, there is an estimated R 270 million (US $60 million) worth of plant medicinal material consumed annually [4]. Moreover, medicinal plants are a majority source of new drugs [5].

South Africa has a unique assemblage of species, and it is the third most biodiverse country globally [6]. It hosts 6% of the global plant species with a 65% rate of endemism, mainly due to its highly varied climate, topography, and geology [6]. South Africa has the fourth highest number of used medicinal plants, after China, India, and Columbia, respectively [7]. However, some of these medicinal plants are under serious threat of extinction from habitat loss and climatic change. A study by Williams et al. [8] indicated that approximately 10% of South Africa's flora are used for traditional medicines, and 3% of these are already on the red list of the International Union for Conservation of Nature (IUCN) and are predicted to be at risk of extinction. Wiersum et al. [9] further give caveats on the decline of medicinal plants in South Africa due to overexploitation, which will subsequently affect biodiversity. Considering South Africa's unique plant species, it is imperative to conserve these kinds of plants by investigating factors that affect their populations. For instance, climate change is likely to have a great negative impact on these medicinal plants.

Like in other countries in Africa, studies in South Africa indicated a considerable climatic change over the last previous five decades [10]. In the country, the mean annual temperatures have risen by nearly 1.5 °C above the recorded global average of 0.65 °C, while the frequency of extreme events has also considerably increased [10]. Particularly, South Africa is vulnerable and sensitive to climate change because of its geographical location and socioeconomic development status. Also, the country is already a warm and arid region that is expected to become even warmer and drier in the future [11]. Moreover, climate change forecasts in South Africa showed that under an intermediate emission scenario (representative concentration pathway: RCP4.5) [12], temperatures will rise by approximately 4 °C by the end of the century [13], while the forecast under the RCP8.5 scenario [14] predicted a temperature increase of up to 6 °C [13]. Likewise, it is forecasted that the amount of rainfall in South Africa will decrease, while the frequency of droughts and dry spells will increase [13]. Climate mainly determines the distribution of species, and a change in climate is highly likely to affect their distribution [15]. Various studies have shown that medicinal plants will decrease in their spatial distribution due to climate change. Evidence of such changes has been documented on plants such as *Schisandra sphenanthera* in China [16] and several plants in Egypt [17]. This will likely be a worldwide problem as global warming will alter several ecosystems by affecting the distribution of plant populations and driving some to extinction [18,19]. In addition, there are predictions that illustrate that some plant relationships with pollinators will be disrupted. For example, Kudo et al. [20] reported that the early onset of spring influences the phenological mismatch between plants and pollinators, resulting in lower seed production due to low pollination service and affecting plant populations. Furthermore, an increase in temperature in areas with cold climates will increase the prevalence of diseases, pests, and pathogens, seriously affecting the plant populations [21].

In addition, climate change is predicted to negatively affect human health. Change in temperatures may expose humans to temperature extremes and poorer nutritional status [22]. In South Africa, the recent rise in temperatures has resulted in an increase in various diseases such as malaria, cholera, and diarrhea [23,24]. Subsequently, the rise in diseases will mean an increase in the use of medicinal plants. Therefore, there is a need to focus on future conservation measures for medicinal plants.

Ecological models have been explored to demonstrate the prediction of the distribution of medicinal plants, with most studies highlighting a decrease in their extent [16]. A recent study in Indonesia indicated that half of the medicinal plants will lose up to 80% of their current spatial distribution [25]. In some parts of Pakistan, the medicinal plant *Tylophora hirsute* will lose its entire habitat by 2050 [26]. However, some studies have reported that some medicinal plants will increase in spatial distribution. For example, You et al. [27] reported that *Rhodiola* species will increase their habitat distribution in Asia. The response to climate change may vary according to the species' resilience and the environment.

The conservation of medicinal plants means conserving biodiversity, ecosystems, and other species within the same habitat. Qian et al. [28] stated that medicinal plants can be used as flagship species to conserve and monitor biodiversity and raise public awareness of conservation strategies. As such, a climate change study related to the conservation planning of medicinal plants in South Africa is imperative. In the current study, species distribution models (i.e., the maximum entropy: MaxEnt) were utilised to investigate the effect of climate change on the future spatial distribution of medicinal plants in South Africa. Species distribution models are used to investigate the relationship between the presence of medicinal plant species and their surrounding environmental variables (characteristics), such as temperature and rainfall, which are then projected under different climate scenarios [29]. Specifically, the current study investigated three medicinal plant species found in South Africa with different ecological requirements, uses, and IUCN status, and thus these were *Aloe ferox* Mill, *Bowiea volubilis* Harv, and *Dioscorea elephantipes* Eng (Table 1). Due to the medicinal values that are mentioned in Table 1, these medicinal plant species are currently in the South African traditional medicine trade markets. This has led to severe harvesting pressure [8], hence the effect of climate change on their geographical distribution is of paramount importance. *A. ferox* is native to Southern Africa and mainly distributed across South Africa and Lesotho [30], while *B. volubilis* is found consistently across the East African region in Kenya to Southern Africa in South Africa [31], and the *D. elephantipes* plant is primarily found in South Africa and Namibia [32]. The study postulates that the investigated plant species will react differently to future climate scenarios because of differences in their spatial occupancy and environmental requirements. The study results will be used to recommend actions towards conservation and local cultivation of the valued plants and efforts to mitigate the effects of climate change.

## 2. Materials and Methods

### 2.1. Medicinal Plant Species

Occurrence records of the plants consisting of geographical positioning system (GPS) coordinates were obtained from the South African National Biodiversity Institute (http://posa.sanbi.org; SANBI, accessed on 7 April 2022) [33] and the Global Biodiversity Information Facility (https://www.gbif.org; GBIF, accessed on 7 April 2022) [34]. A total of 1662, 106, and 142 GPS locations were obtained for *A. ferox* Mill, *B. volubilis* Harv, and *D. elephantipes* Eng, respectively.

**Table 1.** The uses and the International Union for Conservation of Nature (IUCN) status of *Aloe ferox*, *Bowiea volubilis*, and *Dioscorea elephantipes* in South Africa.

| Species Name | Common Name | Uses and Properties | IUCN Status | References |
|---|---|---|---|---|
| *A. ferox* Miller | Cape aloe | Anti-inflammatory, analgesic, wound healing, used as a laxative; relief of arthritis pain, antioxidant, anticancer, antimalarial activities | Least Concern | [35–38] |
| *B. volubilis* Harv | Climbing onion | Purgatives, skin disorders, pains and inflammation, antimicrobial, anti-inflammatory | Vulnerable | [8,31,39–41] |
| *D. elephantipes* Engl | Elephant's foot | Cortisone and contraceptives | Declining | [8,42] |

### 2.2. Climatic Variables for Current and Future Scenarios

In the current study, climatic data were obtained from the WorldClim database (www.worldclim.org/current; accessed on 18 February 2022) [43]. These climatic variables were generated by interpolating the average altitude, temperature, and rainfall data between 1950 to 2000 [44]. The downloaded data had a resolution of 30 s (approximately 1 km$^2$). The current study utilised 19 bioclimatic variables from the WorldClim database (Table 2). These bioclimatic variables are frequently used in investigating species distribution [45–47].

**Table 2.** Bioclimatic variables and their contribution to the maximum entropy (MaxEnt) model experiments for the habitats of *Aloe ferox*, *Bowiea volubilis*, and *Dioscorea elephantipes* in South Africa. The unit of the temperature-based variables is °C, while the unit for the precipitation-based ones is mm.

| Bioclimatic Variable | Code | Contribution | | |
|---|---|---|---|---|
| | | *A. ferox* | *B. volubilis* | *D. elephantipes* |
| Annual Mean Temperature | Bio1 | 1.2 | 0 | 0 |
| Mean diurnal range (Mean of monthly (max temp–min temp)) | Bio2 | 0 | 56.1 | 19.4 |
| Isothermality (Bio2/Bio7) (×100) | Bio3 | 2 | 2.2 | 6.2 |
| Temperature seasonality (standard deviation, ×100) | Bio4 | 1.6 | 0 | 0 |
| Max temperature of the warmest month | Bio5 | 0.6 | 0 | 1.0 |
| Min temperature of the coldest month | Bio6 | 7.0 | 5.1 | 6.3 |
| Temperature annual range (Bio5-Bio6) | Bio7 | 0 | 0 | 0 |
| Mean temperature of wettest quarter | Bio8 | 1.0 | 1.9 | 4.9 |
| Mean temperature of driest quarter | Bio9 | 0.5 | 9.8 | 12.2 |
| Mean temperature of warmest quarter | Bio10 | 0 | 2 | 0 |
| Mean temperature of coldest quarter | Bio11 | 0 | 0 | 0 |
| Annual precipitation | Bio12 | 0 | 0 | 17.4 |
| Precipitation of wettest month | Bio13 | 0 | 0 | 0 |
| Precipitation of driest month | Bio14 | 7.4 | 0 | 0.3 |
| Precipitation seasonality (Coefficient of Variation) | Bio15 | 5.2 | 9.8 | 4.8 |
| Precipitation of wettest quarter | Bio16 | 0 | 0 | 0 |
| Precipitation of driest quarter | Bio17 | 66.3 | 1.5 | 0 |
| Precipitation of warmest quarter | Bio18 | 6.9 | 1.4 | 0 |
| Precipitation of coldest quarter | Bio19 | 0.2 | 10.1 | 27.6 |

This study utilised a mid-term future projection for 2050 and a long-term future projection for 2080, which were obtained from the Consultative Group on International Agricultural Research's (CGIAR) Research Program on Climate Change, Agriculture and Food Security's (CCAFS) climate data archive (http://ccafsclimate.org accessed on 18 February 2022). The future climatic projections used in this study were based on the Coupled Model Intercomparison Project Phase 5 (CMIP5) models (CMIP5) [48]. Previous studies have utilised the CIMP5 models to investigate the effect of climate change on plants [49,50]. The current study specifically compares two future climate scenarios, thus, the representative concentration pathways RCP2.6 and RCP8.5 [44,51]. These Representative Concentration Pathways (RCPs) indicate greenhouse gas concentration trajectories [52]. The RCP2.6 scenario represents that global annual emissions peaked between the years 2010 and 2020, followed by a decline in emissions, while RCP8.5 represents a continuous rise in global emissions [52].

*2.3. Predictive Modelling for Plant Distribution*

The current study utilised the maximum entropy model (MaxEnt version 3.4.3; https://biodiversityinformatics.amnh.org/open_source/MaxEnt/Phillips, accessed on accessed on 8 March 2022) [53], to predict the distribution of the studied medicinal plants. The MaxEnt model employs environmental variables extracted from the presence-only data to predict the distribution of species by finding the probability distribution of the maximum entropy [54]. MaxEnt has been shown to be relatively efficient in modelling species distributions even with small sample sizes [55,56]. The MaxEnt model produces the habitat suitability of the species being investigated, indicating suitability scores (high to low) of the studied species in the study area. Additionally, MaxEnt offers a jackknife tool that calculates the relative importance of each predictor and provides response curves for each predictor variable.

A ten-fold cross-validation method was used to test the performance of the MaxEnt models [57]. In addition, ten replicates were run for each modelling scenario and species, and the results presented herein indicate the average of the replications. The contribution of the individual variables to the models was determined by using the jackknife method from the MaxEnt model. The performance of the models was evaluated by using the area under a receiver operating characteristic (ROC) curve (AUC). The AUC is calculated from the rate of the false positive versus the true positive rate of the model [58,59]. An AUC

value of less than 0.50 indicates a random prediction of the models, while a value closer to 1.0 indicates a better prediction by the model.

Firstly, the prediction of the habitat for each species was modelled using all the 19 bioclimatic variables using the MaxEnt model (Table 2). All the variables that had a greater zero contribution to the models were retained for consecutive model runs. Secondly, the retained variables were imported into an Ecological Niche Modelling (ENM) Tool [60] and subjected to correlation analysis. Pearson's correlation coefficient was computed between two bioclimatic variables at a time (Figure 1). Using highly correlated variables in the models might result in multicollinearity [61]. When two variables had a high correlation (a Pearson's correlation coefficient of greater than 0.8), one variable with a greater contribution in the prior model's run was kept for further modelling [62]. Lastly, the retained variables were then imported into the MaxEnt program for the final running of the models for each studied species. The "10th percentile training presence threshold" was used to define the habitat and non-habitat suitable and non-suitable habitats for the studied medicinal plant species [63]. Additionally, the 'fade-by-clamping' option was employed in MaxEnt, which eliminates severely clamped pixels from the final forecasts, to prevent inaccurate estimates of the appropriate habitat under future temperature scenarios [54]. The products of MaxEnt were then exported to ArcGIS version 10.5 for the final outputs of the map.

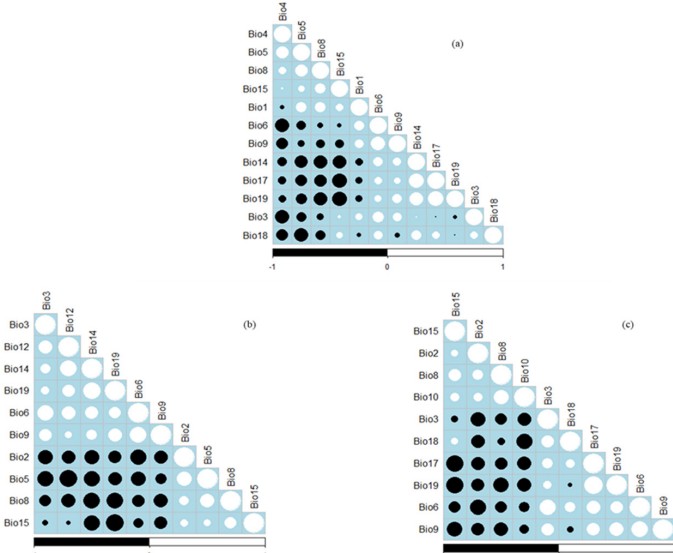

**Figure 1.** Pearson correlation coefficients for the environmental variables retained for the maximum entropy (MaxEnt) distribution modelling of (**a**) *Aloe ferox*, (**b**) *Bowiea volubilis*, and (**c**) *Dioscorea elephantipes*.

## 3. Results

The current distribution of the medicinal plants investigated in this study was mainly affected by different bioclimatic variables (Table 2). The distribution of *Aloe ferox* was mainly affected by the precipitation received in the driest quarter, whereas *B. volubilis* was largely influenced by the mean annual range. *D. elephantipes* was mostly influenced by the amount of precipitation received in the warmest quarter.

The MaxEnt model for *A. ferox* performed very well, with an average AUC of 0.924 and a standard deviation of 0.004. The current distribution of *A. ferox* was mainly in the eastern parts of the country, covering the Eastern Cape, KwaZulu-Natal, and Mpumalanga Provinces (Figures 2a and 3a). The habitat distribution of *A. ferox* was mainly affected by precipitation-based variables, which were the precipitation of the driest month (Bio14) and the precipitation of the warmest quarter (Bio18) (Figure 4a). The future predictions indicated that the distribution of *A. ferox* will reduce in habitat cover, especially in the northeastern parts of the current distribution. The RCP8.5 scenario for 2080 indicated the

most significant loss of *A. ferox* habitats (Figure 3c). However, there will be an overall increase in distribution caused by gained coverage in the southern parts of the country (Table 3). The response curves generated indicated the species' relationship with the bioclimatic variables. The species response curve for *A. ferox* indicated that the plants preferred habitats with the driest month's precipitation ranging from 80 to 200 mm and precipitation in the warmest quarter range of from 0 to 10 mm (Figure 5a,b).

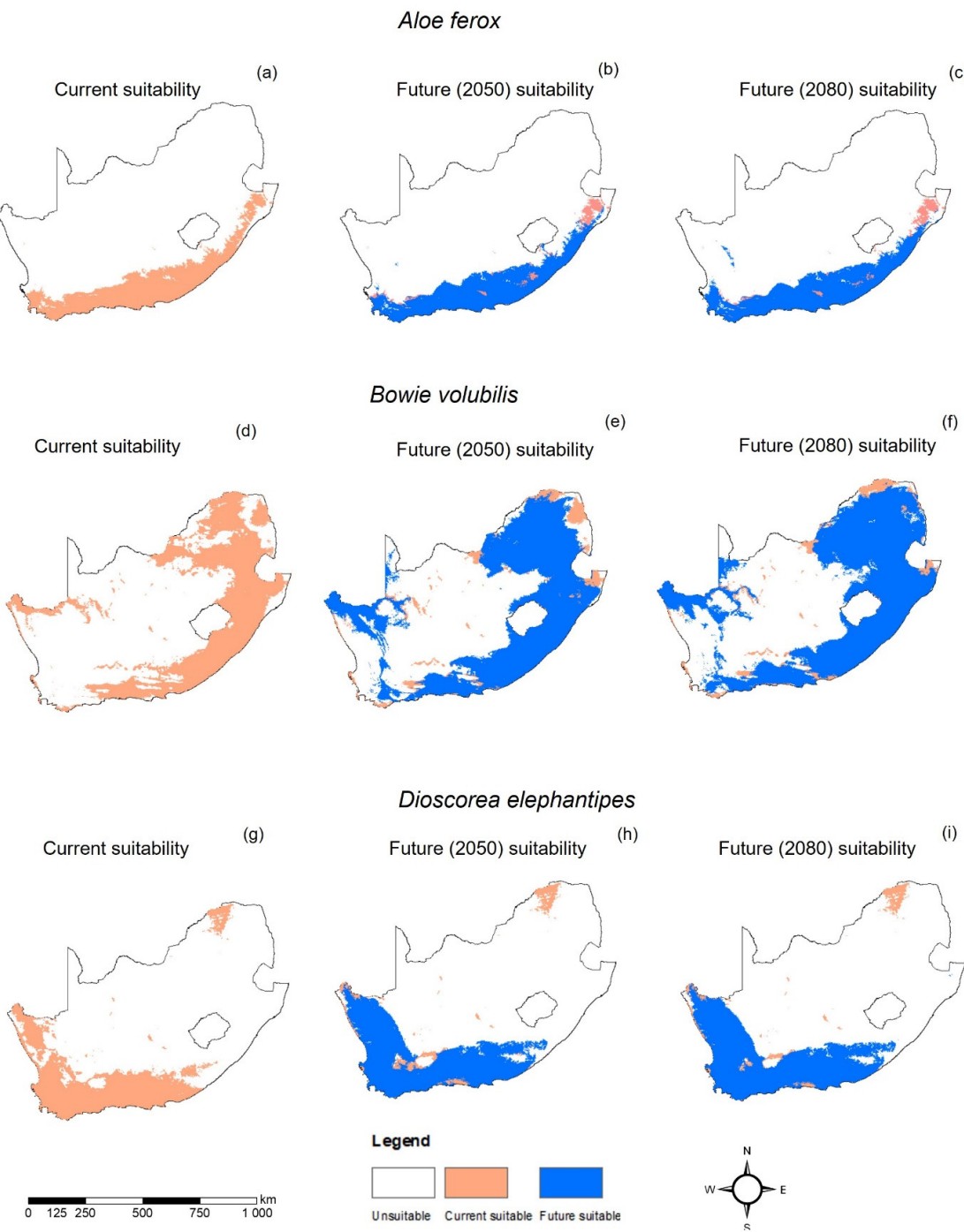

**Figure 2.** Current and projected future change in geographic range of *Aloe ferox* (**a–c**), *Bowiea volubilis* (**d–f**), *and Dioscorea elephantipes* (**g–i**) under the Representative Concentration Pathways (RCP) 2.6 scenario in South Africa.

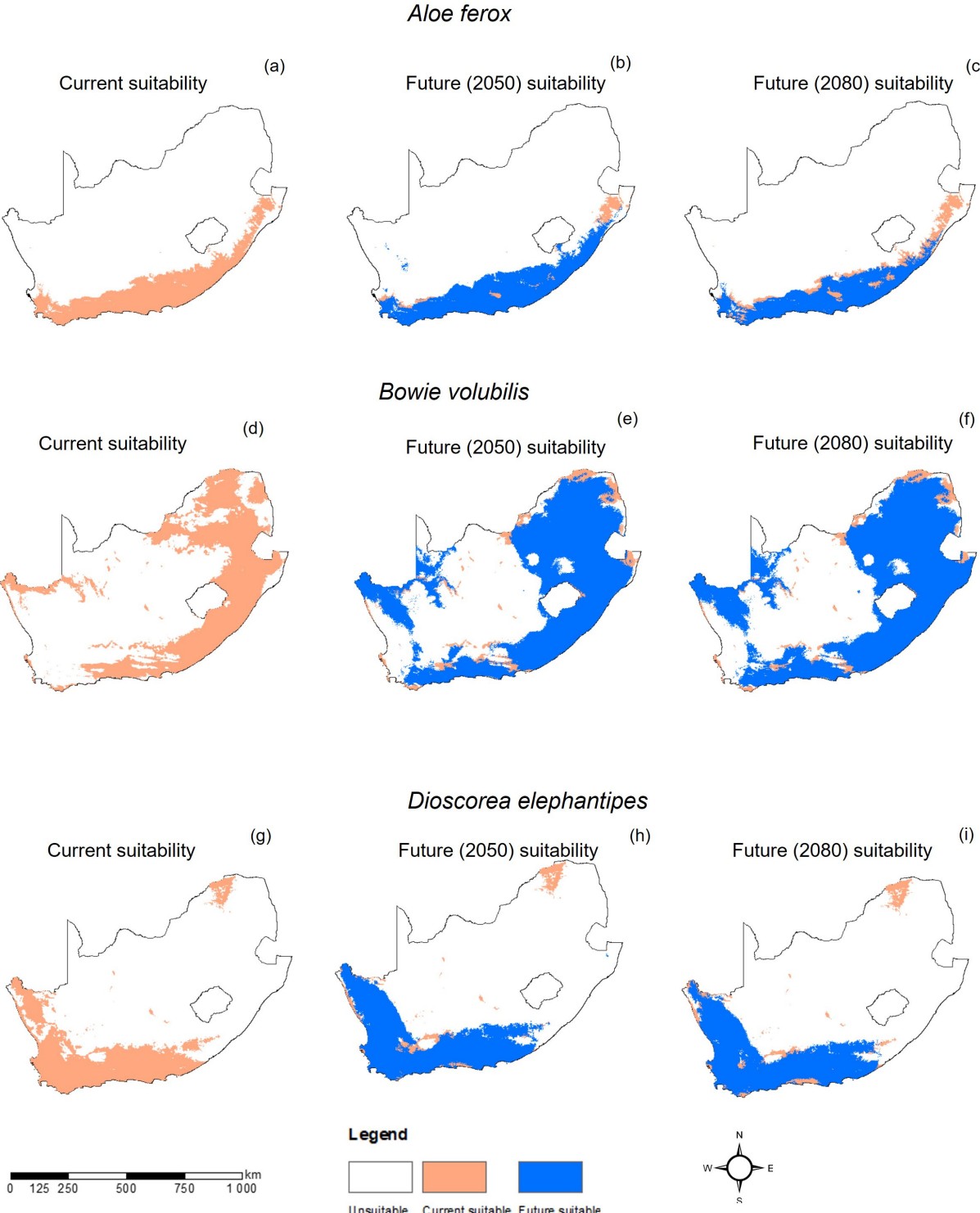

**Figure 3.** Current and projected future change in geographic range of *Aloe ferox* (**a–c**), *Bowiea volubilis* (**d–f**), *and Dioscorea elephantipes* (**g–i**) under the Representative Concentration Pathways (RCP) 8.5 scenario in South Africa.

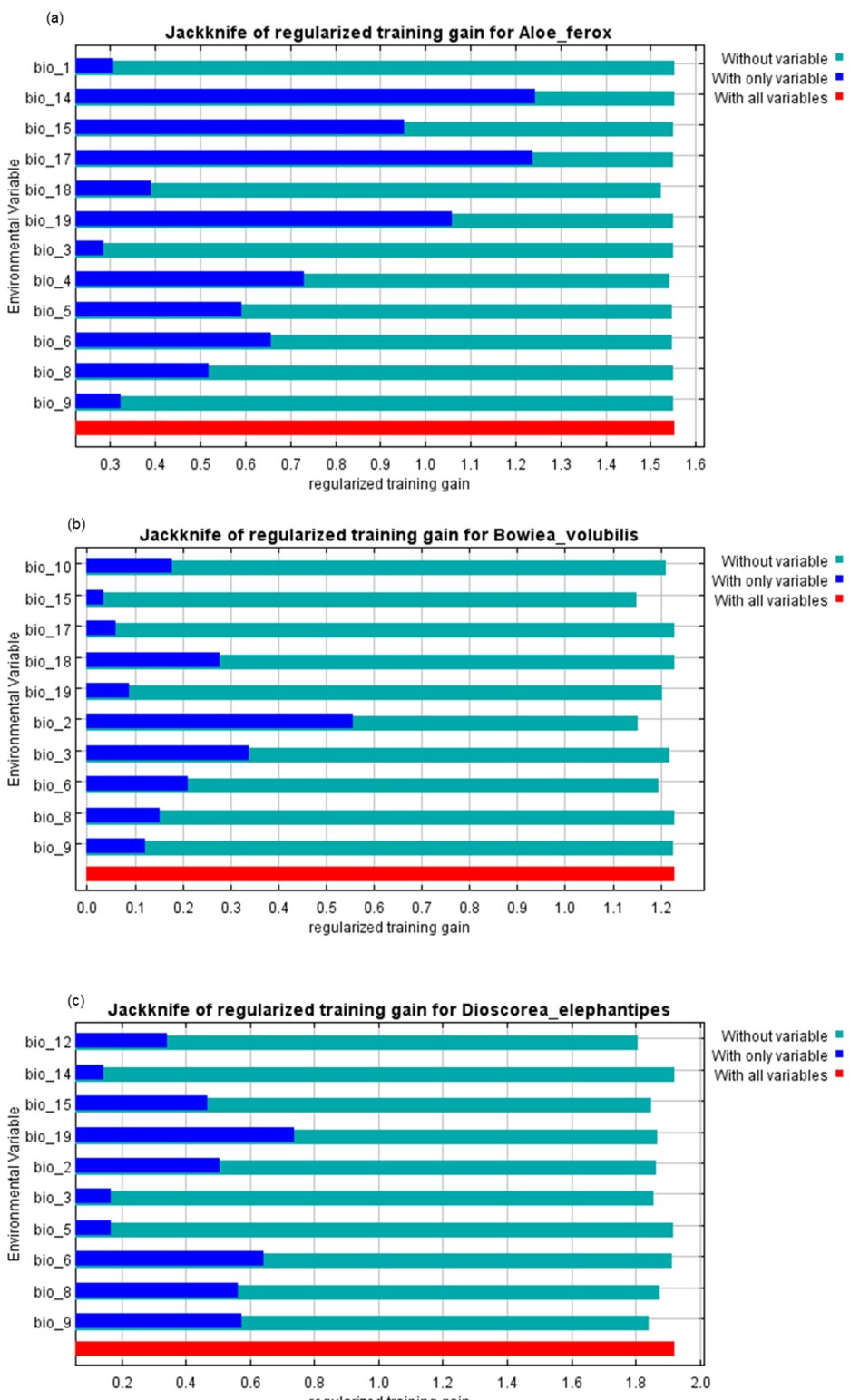

**Figure 4.** Relative predictive power of bioclimatic variables in South Africa based on the jackknife of regularised training gain in the maximum entropy (MaxEnt) models for (**a**) *Aloe ferox*, (**b**) *Bowiea volubilis*, and (**c**) *Dioscorea elephantipes*.

**Table 3.** The maximum entropy (MaxEnt) model prediction of the current and future land cover area (km$^2$) using the representative concentration pathways—RCP2.6 and RCP8.5 scenario models—for 2050 and 2080 in South Africa for *Aloe ferox*, *Bowiea volubilis*, and *Dioscorea elephantipes*.

| Model | *A. ferox* | *B. volubilis* | *D. elephantipes* |
|---|---|---|---|
| Present Coverage | 194,102.52 | 460,788.04 | 211,866.66 |
| RCP2.6 2050 | 197,377.92 | 579,786.05 | 284,663.52 |
| RCP2.6 2080 | 200,958.07 | 594,234.49 | 286,920.43 |
| RCP8.5 2050 | 199,716.27 | 599,473.74 | 266,784.58 |
| RCP8.5 2080 | 203,727.99 | 550,154.24 | 278,999.42 |

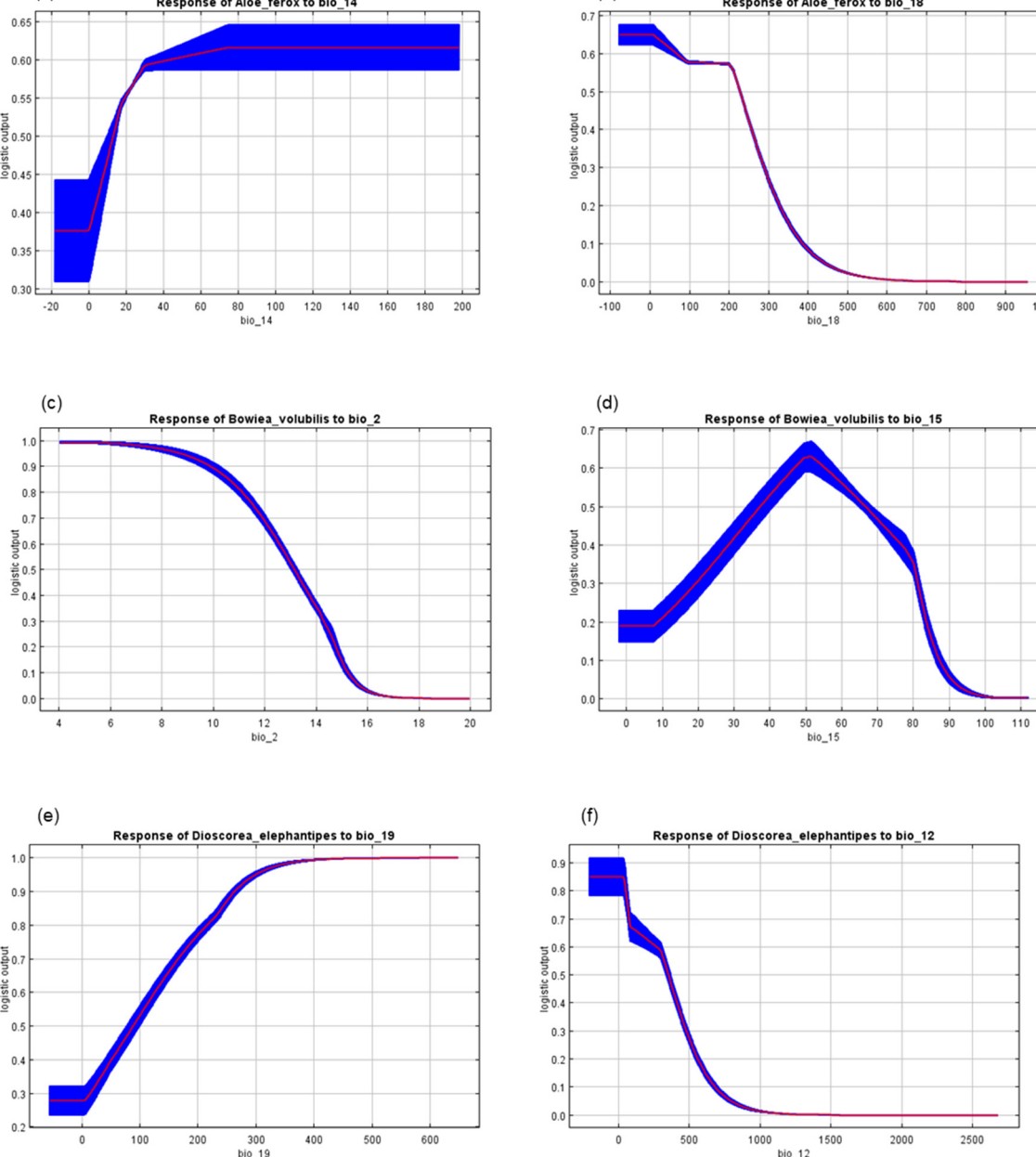

**Figure 5.** Species response curves indicating the relationships among the bioclimatic variables and (**a**,**b**) *Aloe ferox*, (**c**,**d**) *Bowiea volubilis*, and (**e**,**f**) *Dioscorea elephantipes* in South Africa. The red lines indicate the mean of 10 replicates of the maximum entropy (MaxEnt) runs, and the blue lines represent +/− one standard deviation.

The MaxEnt model performance for *B. volubilis* had a mean AUC value of 0.884 with a standard deviation of 0.050. The present distribution of the suitable habitat for *B. volubilis* mainly occurred in the eastern and northern parts of the country (Figures 2a and 3a). Figure 4b indicates that the mean diurnal range (Bio2) and precipitation seasonality (coefficient variation) (Bio15) had the greatest influence on the distribution of *B. volubilis.* This was also evidenced by the mean diurnal range having the highest contribution (56.1%) to the MaxEnt model (Table 2). The future distributions of suitable habitats for *B. volubilis* showed that there will be a slight decrease in coverage in the northern parts of the country covering the Limpopo Province. This was evidenced in both the RCP2.6 (Figure 2e,f) and RCP 8.5 (Figure 3e,f) scenarios. This is also evidenced in Table 3, which highlights an overall increase in coverage. However, there will be a slight increase in distribution into the drier inland parts of the country, noticeably in the Karoo Region in both of the future scenarios, RCP2.6 (Figure 2e,f) and RCP8.5 (Figure 3e,f). The presented response curves show that the probability of the presence of *B. volubilis* decreased as the mean diurnal range increased (Bio2) (Figure 5c), while initially increasing sharply as the precipitation seasonality (Bio15) (Figure 5d).

The MaxEnt model for *D. elephantipes* performed very well with a mean AUC value of 0.944 and with a standard deviation of 0.030. The currently suitable habitats for *D. elephantipes* were mainly predicted to be in the southern and lower western parts of the country, with small coverage in the north (Figures 2a and 3a). The jackknife test indicated that the distribution of *D. elephantipes* was mainly influenced by precipitation of the coldest quarter (Bio19) and annual precipitation (Bio12) and contributed 27.6% and 17.4%, respectively (Figure 4c). Both of the future prediction scenarios indicated that there would be a loss of *D. elephantipes* habitats in the northern parts of the country, while there would be a slight increase in the current habitats of this plant covering the southern parts (Figures 2b,c and 3b,c). There was an overall gain in total land cover by the species in future scenarios (Table 3). The suitability habitat for *D. elephantipes* increased with increasing precipitation of the coldest quarter (Bio19) (Figure 5e); however, it decreased sharply with increasing annual precipitation (Bio12) (Figure 5f).

## 4. Discussion

Climate change has been predicted to affect the geographic distribution of global plants. Predicting the effect of climate change on species' habitats gives guidance on the expected future spatial distribution changes [64]. The current study highlighted some geographical shifts of habitats for important medicinal plants in South Africa.

The current study utilised MaxEnt modelling, which is a species distribution modelling method. The AUC values obtained for modelling the medicinal plants were 0.924, 0.884, and 0.944 for *A. ferox*, *B. volubilis*, and *D. elephantipes*, respectively. Pearce et al. [65] stated that a model with an AUC value of 0.75 is considered to be reliably accurate. Therefore, the AUC values obtained in the present study are within the acceptable range of the species distribution model's accuracy. However, some species distribution modelling algorithms such as MaxEnt are data-driven and considerably rely on the parameter settings; hence, they can result in biased estimations, incorrect inference, and poor performance when upscaled to new conditions [66]. This is primarily due to the fact that species distribution models are intrinsically dependent on the prevalence of the species' occurrence observations, and, consequently, they could introduce statistical artefacts into the estimations of the predictive accuracy [67,68]. Allouche et al. [69] reported that different spatial distribution models can result in different distribution predictions for the same species under the same conditions. While other studies, such as those by Araújo, et al. [70] and Grenouillet, et al. [71], advocated for the use of ensemble methods, which have been shown to reduce model uncertainty by averaging the results from various species distribution approaches. However, caution needs to be observed as this may result in further inaccuracies. For instance, Crimmins et al. [72] demonstrated that ensemble methods may not be the rightful approach for predicting species distributions under future climatic conditions when compared to the use of a single species distribution model.

These results indicated that *A. ferox* is affected mainly by precipitation-based variables, with droughts predicted to be prevalent in the future [73], which will most likely affect the habitat of *A. ferox*. For example, in East Africa, the RCP2.6 and RCP8.5 future scenarios have been characterised by an increase in drought areas of up to 16% and 54%, respectively [74]. This corroborates the results of the current study showing a reduction in the habitat of *A. ferox* the most in some areas, most likely due to future droughts. The habitat of a similar species, *A. vera,* has also been predicted to shrink due to climate change in the near future [75]. In addition, it has been predicted that the loss of the *A. ferox* species will be exacerbated in the Eastern Cape Province in the next coming 100 years, mainly due to herbivory [76,77]. These authors showed that local extinctions are likely to be caused by herbivory, especially on the younger plants, creating a demographic bottleneck. Therefore, a combination of these factors and harvesting pressure from humans is likely to drive the extinction of this species at a faster rate.

Furthermore, the present study indicated that there will be a reduction in the suitable habitats of *B. volubilis* and *D. elephantipes*, especially in the northern parts of the country. This was explicitly shown for the habitat distribution for *D. elephantipes,* where there was a total shrinkage of the habitat in the northern parts of the country. Research has shown that the species are now shifting towards the poles because the areas closer to the equator have and will further experience higher temperatures [78–80]. The results of the current study are in line with Kapwata et al. [81], who predicted that by 2088 to 2099 there will be a 4°C rise in the average temperatures in the southern parts of South Africa, while a 6 °C increase will be experienced in the northern, western, and central parts of the country. Moreover, this study highlights the variable precipitation in the coldest quarter, and other temperature-related variables contributed just above 70% in influencing the habitat distribution of *D. elephantipes.* Indeed, other studies have reported similar findings; for example, in Southwestern Australia, up to two-thirds of the plant species will decline and lose up to 25% of their habitats by 2080 [82]. Similarly, ecological models are effective in predicting medicinal plant distributions, with both reports of increase and decrease in their distribution [83–85]. However, unlike the species distribution models, ecological models require a number of parameters that might not always be easily available. Moreover, species distribution models relate the species occurrence observation to a set of readily available environmental variables and provide insights on which variable(s) has the most influence on and contribution to the occurrence and distribution of the species of interest.

Investigating the effect of climate change on species distribution is complex because there are chances of both increases and decreases in the habitat distribution happening concurrently. This was shown in the distribution of the studied species. Similar findings have been reported for species such as *Vincetoxicum arnottianum* [26] and *A. dichotoma* [45]. The addition of new habitats in future scenarios highlights the species' abilities in adapting to the new climate. This is likely through phenological or physiological adaptations as survival tactics for the species.

The study highlighted that some habitats will totally be lost in some parts of the country. This will severely affect the local people, as they strongly rely on these medicinal plants. Hamilton [5] stated that a total of 80% of the population in developing countries rely heavily on medicinal plants. Therefore, the conservation of these species should be encouraged to include ex situ conservation to counter the effects of climate change and the creation of protected areas for in situ conservation to avoid total local extinction of the species [86]. For example, the Pepperbark tree (*Warburgia salutaris*), which is one of the most highly prized medicinal plants, was once considered extinct in Zimbabwe but was reintroduced into the country from South Africa through ex situ conservation [87].

Although the models used in the current study had great validation of AUC values, there is still a level of uncertainty regarding the distribution of the species. Firstly, the study utilised bioclimatic variables, while other factors, such as the species dispersal mode, land cover changes, and anthropogenic factors, especially harvesting, were not considered in the medicinal plant habitat modelling experiments. The inclusion of these factors has

the potential of driving more reliable species distribution modelling output. However, the species distribution models developed in the present study give critical guidance for adapting strategies and policies to counter the effects of climate change.

Our study highlighted the potential shifts in the distribution of medicinal plants in South Africa. The results of the present study could guide the development of effective and efficient policies and strategies for managing and conserving medicinal plants in South Africa. Additionally, it is necessary to investigate how climate change affects the phytochemical composition, as changes in environmental conditions will likely alter the medicinal phytochemical concentrations and possibly render them toxic for human consumption [88,89].

## 5. Conclusions

Considering the results from our current study, climate change will negatively influence the geo- and temporal distribution, abundance, and bio(geo) diversity of medicinal plant species in South Africa. The study illustrated that the habitats of species such as *D. elephantipes,* which are mostly affected by precipitation in the higher latitudes, will most likely shift due to climate change. Therefore, conservation actions, such as the establishment of protected areas, planting such plant species, and using alternative energy sources, are highly needed to prevent such valuable medicinal plant species from becoming extinct. Furthermore, other various measures need to be put into effect to protect the local pharmaceutical and income resources, which include medicinal plant species. This can include incorporating future forecast climate scenarios into current conservation initiatives and policies, while, on the other hand, the climate will continue to change, largely as a result of environmentally unfriendly anthropogenic activities and policies that govern them. Therefore, initiatives advocating for species' protection and mitigating the effects of climate change should be prioritised.

**Author Contributions:** Conceptualization, T.T., O.M. and E.M.A.-R.; methodology, T.T.; formal analysis, T.T.; writing—original draft preparation, T.T.; writing—review and editing, T.T., O.M. and E.M.A.-R.; supervision, O.M. and E.M.A.-R.; funding acquisition, O.M. All authors have read and agreed to the published version of the manuscript.

**Funding:** This research was supported by the National Research Foundation (NRF) South African Research Chairs Initiative (SARChI) chair in Land-Use Planning and Management (Grant number: 84157).

**Institutional Review Board Statement:** Not applicable.

**Informed Consent Statement:** Not applicable.

**Data Availability Statement:** Data available on request.

**Conflicts of Interest:** The authors declare no conflict of interest.

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
