# Peer review of "Predicting the Geographical Distribution Shift of Medicinal Plants in South Africa Due to Climate Change"

_conservation, doi:10.3390/conservation2040045_

Round 1
Reviewer 1 Report
On the basis of careful study of the manuscript number 1937854, it results well written, the experiments well designed and the topic relevant to be published in Conservation.
Only few comments:
It would be interesting to include if the species are endemic or, otherwise, their worldwide distribution.
Line 486: delete reference 73
Author Response
Please see the attachement.

Reviewer 2 Report
Title: Predicting the geographical distribution shift of medicinal plants in South Africa due to climate change
This topic is important and best fit in this journal. But I have some suggestions before the acceptance of this article. Authors should improve:
Comments
1. Why did the authors choose specifically three medicinal plants (Aloe ferox, Bowiea volubilis, and Dioscorea elephantipes) for the present study? Kindly justify in the introduction section.
2. In the introduction, the authors should address the climatic variation of the last few decades in South Africa. Kindly add this information in the appropriate place.
3. In the materials and method section, the authors should indicate the year and season in which this study was conducted.
4. All the three plants (Aloe ferox, Bowiea volubilis, and Dioscorea elephantipes) should be abbreviated (A. ferox, B. volubilis, and D. elephantipes) throughout the manuscript except in abstract.
5. There are some grammatical errors in the manuscript. The authors should thoroughly check the manuscript and correct it.
6. In the conclusion section, the authors should add an in-depth opinion and future prospectus on the conservation of plant species based on this present study.
7. In reference section Ref. No. 61, the scientific name of the plant ‘Aloe vera’ should be in italic
8. In reference section Ref. No. 71, the scientific name of the plant ‘Senecio jacobaea, Senecio aquaticus’ should be in italic
9. Reference section may check thoroughly and modify to Journal format.
Recommendation
The manuscript needs revision
Reviewer 3 Report
Reviewer’s comments
The manuscript entitled “Predicting the geographical distribution shift of medicinal plants in South Africa due to climate change by Thulani Tshabalala, Onisimo Mutanga, and Elfatih M. Abdel-Rahman, talks about the effect of climate change on three medicinal plants (Aloe ferox, Bowiea volubilis and Dioscorea elephantipes) distribution in South Africa. Further, a species distribution model (i.e., the maximum entropy: MaxEnt) was used to investigate the effect of climate change on the future spatial distribution of medicinal plants in South Africa.
Medicinal plants and resources are valuable in the traditional medicine system and need to be conserved on a priority basis. The study provides interesting findings on the prediction of the decline or expansion of plant species in response to fluctuating climatic conditions in S. Africa.
Specific comments:
Studies have suggested that Ecological models are effective predictors of the distribution of medicinal plants, with both reports of increase and decrease in the distribution. However, the species distribution model in the present study also reports variations in the predictions of plant species distribution. How this study and model prediction is better than other prediction models? Discuss.
Compare and contrast how the use of species distribution models is better than other prediction models.
Although the study is an interesting one providing a platform for future predictions of endangered plant species, most of the references cited are a bit old. New emerging studies in this direction would greatly benefit the paper, particularly the discussion/conclusion.
The reviewer suggests minor revisions of the manuscript before acceptance.

Round 2
Reviewer 2 Report
The author has addressed my previous comments successfully .
The author should correct scientific name of the plants should be in italic in reference sections.
